# Bilingual Spatial Cognition: Spatial Cue Use in Bilinguals and Monolinguals

**DOI:** 10.3390/brainsci14020134

**Published:** 2024-01-27

**Authors:** Anna Tyborowska, Joost Wegman, Gabriele Janzen

**Affiliations:** 1Behavioural Science Institute, Radboud University Nijmegen, 6500 HE Nijmegen, The Netherlands; anna.tyborowska@ru.nl (A.T.); joostwegman@gmail.com (J.W.); 2Donders Institute for Brain, Cognition and Behaviour, Radboud University Nijmegen, 6500 HD Nijmegen, The Netherlands

**Keywords:** bilingualism, executive control, navigation, spatial cues, fMRI

## Abstract

Structural plasticity changes and functional differences in executive control tasks have been reported in bilinguals compared to monolinguals, supporting a proposed bilingual ‘advantage’ in executive control functions (e.g., task switching) due to continual usage of control mechanisms that inhibit one of the coexisting languages. However, it remains unknown whether these differences are also apparent in the spatial domain. The present fMRI study explores the use of spatial cues in 15 bilinguals and 14 monolinguals while navigating in an open-field virtual environment. In each trial, participants had to navigate towards a target object that was visible during encoding but hidden in retrieval. An extensive network was activated in bilinguals compared to monolinguals in the encoding and retrieval phase. During encoding, bilinguals activated the right temporal and left parietal regions (object trials) and left inferior frontal, precentral, and lingual regions more than monolinguals. During retrieval, the same contrasts activated the left caudate nucleus and the right dorsolateral prefrontal cortex (DLPFC), the left parahippocampal gyrus, as well as caudate regions. These results suggest that bilinguals may recruit neural networks known to subserve not only executive control processes but also spatial strategies.

## 1. Introduction

In recent decades, bilingualism has quickly become a worldwide phenomenon spreading as a result of modern globalization. Many have argued that the use of a second language engages domain-general executive functions, leading to a ‘bilingual advantage’ [1,2,3], whereas others find no evidence for a bilingual advantage in executive task switching [4,5,6,7,8]. These studies, however, do not tap into the complexity of underlying cognitive processes in more real-life situations, such as navigating through known and unknown spatial environments. Successful wayfinding is a critical ability that constitutes an essential part of our daily lives. However, it remains unclear whether the bilingual ‘advantage’ is also present in the spatial domain. To investigate this, we propose a novel approach utilizing a navigation task simulating real-world settings.

Bilinguals have the remarkable task of managing both of their languages during production and comprehension processes. As a result, a bilingual individual can be considered a “speaker-hearer” with a unique linguistic composition due to the coexistence and constant interaction of the two languages [9]. As such, the bilingual in Grosjean’s (2008) view is not the sum of two monolinguals but rather an individual with one qualitatively different linguistic system. Stemming from this assumption is the functional definition of bilingualism as the regular use of two languages in everyday life.

The necessity of managing two competing languages requires the continuous use of a control mechanism which inhibits one of the languages [10]. This mechanism is believed to also impact a general cognitive domain involving executive functions [11]. Studies show that bilinguals, compared to monolinguals, have an advantage in nonverbal tasks recruiting inhibition, task switching, and selective attention processes [12,13,14,15,16]. This bilingual benefit is not always observed. There is an ongoing controversy in the field regarding the bilingual advantage in executive control [1,2,3,4,5,6,7,8].

Neuroimaging results additionally point to structural and functional differences between bilinguals and monolinguals and suggest a neural signature of bilingualism with respect to, e.g., syntactic sentence judgment [17], overt word production [18,19], and language control mechanism [20]. The latter points to activation of a widespread network, including the anterior cingulate cortex (ACC), left prefrontal cortex, left caudate and bilateral supramarginal gyri regions within the executive control network important for attention and cognitive control [21,22]. Structural differences in bilinguals have been shown for higher grey matter (GM) density in the left inferior parietal cortex [23], which correlated positively with second language proficiency. Older adult bilinguals were reported to have higher white matter integrity in the corpus callosum and superior and inferior longitudinal fasciculi, as well as enhanced anterior-to-posterior frontal connectivity [24]. These studies imply that environmental demands and experience can change brain structure, in this instance, a lifetime of enhanced cognitive control due to being bilingual.

Furthermore, studies investigating the neural correlates of nonverbal cognitive control suggest that bilinguals activate different networks for conflict resolution. Bilingualism has also been shown to tune the ACC for conflict monitoring or resolution [25]. While performing a flanker task, bilinguals were shown to activate an extensive network, including bilateral frontal, temporal, and subcortical regions, in incongruent trials (requiring control), while monolinguals recruited a spatially restricted network (including the left temporal pole and left superior parietal lobe) [26]. Garbin and colleagues [27] go further to suggest that bilinguals recruit brain regions involved in language control when performing nonlinguistic cognitive tasks (e.g., task switching). They show that the lower the switch cost, the more the left inferior frontal cortex is activated in bilinguals.

A few behavioral studies investigating spatial cognition also suggest differences in bilingual and monolingual individuals in nonverbal visuospatial tasks [28,29,30,31,32]. McLeay [29] tested participants on diagrams of knotted and unknotted ropes that had to be compared at varying orientations and found that bilinguals had shorter RTs than monolinguals. Another study reported performance differences between bilinguals and monolinguals in a route-learning task [30] and found that classical gender differences were reversed for bilinguals. Bilingual women were faster in learning the required route and made fewer errors than bilingual men. These studies, however, do not directly address navigational abilities and how bilinguals may benefit in this respect.

Behavioral and neuroimaging studies on spatial navigation have identified individual differences in wayfinding and spatial cue use. Spatial strategies range from egocentric (object-based) to allocentric (world-based) and are likely used based on a combination of individual preferences [33,34,35,36,37,38,39,40] and task demands [41]. Individual differences in strategy use and performance have been observed not only on the behavioral level but on the neural level as well [42,43]. For example, in a study by Baumann and colleagues [44], both encoding and retrieval activated the caudate nucleus, parahippocampal gyrus, and parietal cortex, but activity in these areas was modulated by the participants’ performance. Successful encoding was associated with activity in the right hippocampus and bilateral parahippocampal gyrus, while retrieval of relevant spatial representations was linked with activity in the left hippocampus and bilateral striatum [45,46]. Other studies have also shown a human neural navigation mechanism that denotes destination [47] and automatically marks relevant information during route learning [48,49]. Finally, differential patterns of activation have been shown for good and bad navigators [44,50], as well as differences in functional connectivity [51].

The hippocampus, caudate nucleus and parietal cortex are differentially recruited based on the specific use of spatial cues. The hippocampus and caudate nucleus have been shown to underlie survey (world-based) and route (observer-based) strategies, respectively [43]. These differences are also reflected in hippocampal and caudate GM volumes [52], as well as fractional anisotropy (FA) values in the right hippocampus that correlate positively with cognitive map creation and efficiency in reorienting [53]. Additional support for hippocampal involvement in cognitive maps comes from [54] for boundary-related landmark learning and memory, whereas the caudate nucleus is suggested to underlie landmark-related learning. Of note, [55] dissociate between the right and left hippocampus for allocentric (map-based navigation) and egocentric representations (sequential organization of the environment). Finally, the parietal cortex has been associated with both allocentric [56] and egocentric [47,57] spatial processing, mental navigation [58], and the marking of perceived heading direction [47,59], among other spatial processes [60].

In the current fMRI study, we aim to investigate the extent of the bilingual executive control advantage in a navigation task requiring the use of different spatial cues. The involvement of executive control has been implicated in spatial abilities such as spatial visualization and spatial relations [61]. Prefrontal cortex GM volume has also been associated with learning performance in a virtual Morris water maze task [62], and lesions to these regions have been reported to severely impair navigational abilities [63]. As a result, we expect bilinguals to have an advantage in tasks recruiting such spatial processes. In particular, we want to examine if bilinguals are more proficient in using spatial cues for navigation and whether the ‘bilingual effect’ also holds for spatial cue switching. We expect that bilinguals will show performance differences compared to monolinguals in using spatial cues for navigation. In regard to the neural correlates of spatial cue use, we expect bilinguals to differentially activate an extensive network of regions known to underlie spatial cognition (i.e., hippocampus, parahippocampal gyrus, caudate nucleus, parietal cortex) as well as the executive control network (inferior frontal regions, ACC).

## 2. Materials and Methods

### 2.1. Participants

Thirty-two healthy right-handed adults participated in the study, who indicated their handedness via self-report. Data from three participants were excluded due to structural abnormalities or large movement artifacts that significantly distorted the fMRI signal. Twenty-nine participants were included in the final sample—15 bilinguals (mean age = 22.93, SD = 2.87) and 14 monolinguals (mean age = 23.9, SD = 6.05); the groups did not differ in age, t(27), *p* = 0.56. The proportion of men and women was balanced between groups—there were 5 women and 10 men in the bilingual group and 5 women and 9 men in the monolingual group. All bilinguals were native Dutch speakers highly fluent in English, with the exception of one participant who considered both English and Dutch as his native language. The monolinguals were native English speakers who were not functionally fluent in any other language. All participants were told that the study investigated the interaction between language background and spatial cognition. They were not informed before testing that their bilingualism/monolingualism was the subject of the study in order to eliminate any prior expectations they may have had.

The education level was matched for both groups, with all participants being university students or higher. Due to the nature of the experimental task, participants rated their gaming frequency on the following scale: never (1), once (2), less than once a year (3), less than once a month (4), less than once a week (5), several times per week (6), every day (7). The groups did not differ in the amount of time they spent playing 2D video games, t(27) = 1.23, *p* = 0.226. Bilinguals reported playing video games from less than once a month to less than once a week (M = 4.53, SD = 1.25) and monolinguals less than once a month (M = 3.93, SD = 1.39). The same was true for both groups concerning 3D video games (bilinguals: M = 4.20, SD = 1.15; monolinguals: M = 3.57, SD = 1.40), t(27)= 1.32, *p* = 0.195. All participants gave informed consent prior to testing according to a protocol approved by the local ethics committee (CMO region Arnhem–Nijmegen, The Netherlands, approval code CMO2001/095, December 2010) and received payment or course credit for taking part in the experiment.

#### Language Background

All participants were assessed according to their language background. Bilingual participants filled out an adopted online version of the L2 Language History Questionnaire (Version 2.0) [64]. Monolingual participants also answered a subset of questions taken from this questionnaire to ensure that their contact with another language was no more than minimal. Bilinguals were additionally administered the English version of DIALANG, a standardized Common European Framework language diagnosis system for assessing language proficiency (administered online from the Lancaster University server, http://www.lancs.ac.uk/researchenterprise/dialang/about.htm (first accessed on 7 April 2011)).

All bilinguals reported using English often to daily and rated their general language fluency at least as good to very good. Answers from the L2 Language History Questionnaire were assessed for more detailed information regarding their language background. On a self-rated English proficiency scale from 1 to 7 (very poor, poor, fair, functional, good, very good, nativelike), bilinguals reported their English abilities in reading, writing, speaking, and listening on average as very good (reading: M = 6.47, SD = 0.52; writing: M = 5.87, SD = 0.83; speaking: M = 5.73, SD = 0.70; listening: M = 6.27, SD = 0.46). The average age that participants reported speaking in English was 9.37 (SD = 3.47); this was similar for reading at the age of 9.47 (SD = 2.80) but slightly older for writing at age 10.2 (SD = 2.98). Rated on a scale from 0 to 5 corresponding to 0%, 25%, 50%, 75%, and 100% of daily language use (both spoken and written), bilinguals reported using English on average 25 percent each day (M = 1, SD = 0.85) and Dutch more than 50 percent (M = 2.40, SD = 1.18). All participants were required to use both English and Dutch at university, with the exception of two individuals who used either solely English or Dutch. Finally, the bilingual group also reported knowing at least one other language besides English (with the maximum being 4 other languages by one participant); the average number of other languages was 2.33. Bilinguals were additionally assessed according to their English proficiency with a placement test and the grammar subtest of the DIALANG. Out of a possible 1000 points in the placement test, they obtained an average score of 918.4 (SD = 114.6). In the grammar test, all participants were placed at the C level, which is the highest and denotes a proficient language user (Council of Europe) [65].

All monolinguals met the following criteria: 1. no functional knowledge of another language and 2. no daily use of a known second language. Out of 14 monolinguals, 11 reported rudimentary knowledge of a second language (e.g., from a foreign language course taken at school or due to living in the Netherlands). These participants answered two questions regarding language fluency and frequency of use taken from the L2 Language History Questionnaire. On a self-rated general language fluency scale from 1 to 7 (very poor to nativelike), none of the participants judged this other language beyond 3 (equivalent to fair; with 4 being functional fluency). On average, monolingual participants rated their fluency as 1.9 (corresponding to poor on the scale).

### 2.2. Materials

#### 2.2.1. Spatial Questionnaires

Each participant filled out a set of three spatial questionnaires, on which they had to self-rate their environmental spatial abilities, navigation strategies, and level of spatial anxiety. Because individual differences have been observed in relation to spatial abilities [66], it was important to take this into consideration when analyzing group differences in the experimental task. All questionnaires were administered on a computer in an online version.

##### Santa Barbara Sense of Direction (SBSOD) Questionnaire

The SBSOD [67] is a self-report measure of spatial and navigational abilities, preferences, and experiences. Sense of direction is related to updating one’s location in space during self-motion. Previous research suggests correlations between sense of direction measure with the SBSOD and performance in large-scale real and virtual environments [66,68]. The questionnaire consists of 15 items that need to be rated on a 7-point Likert scale (1—strongly agree, 7—strongly disagree). Approximately half the items are phrased in a positive way and half in a negative way. An example of an item stated positively is, “I am very good at giving direction,” and negatively, “I don’t have a very good ‘mental map’ of my environment”. A higher rating indicates a better sense of direction (meaning that scoring of positively stated items is reversed).

##### Wayfinding Strategy Questionnaire

The Wayfinding Strategy Questionnaire [69] measures the extent to which two types of strategies are used to find one’s way to a new location. It has been shown that people vary in the extent to which they use either strategy, both for indoor and outdoor wayfinding [33,69], and flexibility in using the strategies has been correlated with good sense of direction abilities [34]. The questionnaire consists of 14 items describing either a survey strategy (9 items) or a route strategy (5 items). A survey strategy involves tracking the relationship between one’s own position and global reference points (e.g., “I kept track of where I was in relationship to the sun (or moon) in the sky as I went.”). On the other hand, a route strategy makes use of step-by-step directions referring to local landmarks (e.g., “Before starting, I asked for directions telling me how many streets to pass before making each turn”). The items are rated on a 5-point Likert scale depending on how typical a strategy is considered for the individual (1—not at all typical of me, 5—very typical of me). The scale results in two measures—a separate score for the extent to which survey strategies are used and for route strategies.

##### Spatial Anxiety Scale

The Spatial Anxiety Scale [69] measures the level of anxiety in situations requiring spatial and navigational skills. Previous studies have shown a correlation between higher spatial anxiety and more navigation errors [70] and worse sense of direction performance [33]. The questionnaire has 8 items related to indoor or outdoor wayfinding (e.g., “Finding my way back to a familiar area after realizing I have made a wrong turn and become lost while traveling”) that need to be rated on a scale from 1 (not at all anxious) to 5 (very anxious) pertaining to the amount of anxiety the participant would feel in each situation. As a result, a high score indicates a high level of spatial anxiety.

#### 2.2.2. Mental Rotation Test (MRT)

The MRT [71] is a paper and pencil test of three-dimensional spatial visualization that measures the ability to mentally rotate figures. The test is based on research by Shepard and Meltzer [72] and has been shown to produce a wide range of individual differences. The test consists of 20 problems in which it is necessary to match a target figure to a set of four test figures. The participants’ task is to indicate which two of the test figures are rotations of the target figure as quickly and accurately as possible. The test consists of a practice part and two test sections, each with 10 items. The participants are given 3 min to complete each section. The score reflects the number of correctly completed problems (both test figures in each problem need to be correctly identified).

#### 2.2.3. Eight-Arm Task [43]

The task was a modified version of the eight-arm radial maze virtual environment navigation task used by Iaria and colleagues [43], which in turn was based on the principle of the Morriswater task [73] (see [74] for a detailed description). The three-dimensional virtual environment contained an eight-arm radial maze surrounded by a mountainous landscape with two trees and a sunset in the background. Participants entered the maze from the central location, from which the maze’s arms protruded. At the end of each arm was a wooden box that could be picked up. Each trial consisted of two parts. In the first part, four of the arms were blocked. Participants were instructed to enter the available arms and retrieve the boxes. In the second part, all arms of the maze were available, and participants had to find the remaining four boxes. This meant that they had to enter the arms that had previously been blocked and consequently remember, which arms they had already visited. Participants completed three such trials with the fourth trial being a probe trial. In the second part of this last trial, the wall around the maze was raised to conceal the entire environment beyond, leaving no landmarks visible. If participants were using a spatial strategy involving the landmarks in the environment, they would make more errors in finding the boxes in this trial.

The task was programmed in Blender, an open-source 3D content creation suite (The Blender Foundation Amsterdam, The Netherlands; www.blender.org (accessed 29 March 2011). The sequence of the available arms in the first part was changed from trial to trial and, in accordance with Iaria and colleagues [43], the order of the trials was kept constant across participants. The trial continued until all objects had been picked up (part 1) or found (part 2). Entering an arm without a box in the second part of the trial was counted as an error.

#### 2.2.4. Flanker Task

The flanker task was used to verify that the bilingual group in our experiment could replicate results from previous studies (e.g., [1,13]) showing a bilingual executive control advantage. Participants saw a row of white symbols in the middle of a black screen and were instructed to respond as quickly as possible and as accurately as they could to the middle chevron, which was pointed either to the left or right. In neutral trials, the flanking stimuli were horizontal lines that provided no interfering or facilitating information to the target chevron but had the same number and approximate size of items as other non-baseline trials. In congruent trials, the target chevron was flanked by two chevrons on each side that pointed in the same direction as the target. In incongruent trials, the flanking chevrons pointed in the opposite direction to the one indicated by the target. These trials induced conflict in making a response due to the interfering effect of the distracter flankers. The amount of experienced conflict is measured by the difference in RT between congruent and incongruent trials. A total of 96 trials were presented per condition.

The experiment was programmed using Presentation software (Version 14.9). The behavioral session started with 12 practice trials so that participants could familiarize themselves with the task. The testing session was divided into 6 blocks, each block consisting of 48 trials (16 trials for each condition, with half the target chevrons pointing left and half right) that were randomly intermixed. The participants could begin each consecutive block at their own pace. A fixation cross was presented for 400 ms before the start of each trial. Participants were required to respond with the left and right arrow keys on a computer keyboard with two fingers of their right hand. After each response, the task continued on to the next trial. The maximum allocated duration of a response was set to 1700 ms, after which time the next trial was automatically presented. Response time and accuracy were measured; trials with incorrect responses were excluded from further analyses.

#### 2.2.5. Objects in Space (OiS) Navigation Task (fMRI Task)

Participants performed an open field navigation task [74] in a virtual environment similar to the one used by [44]. Each trial consisted of an encoding and retrieval phase in which participants had to navigate toward a target that was visible during encoding but hidden in retrieval (Figure 1). In the encoding phase of the trial, participants entered an environment that contained three colored columns and a yellow pyramid (the target). An implicit sun (not visible in the environment) cast a shadow off each column. The participants were instructed to navigate towards the target within a limited amount of time (10 s) and remember its location in the environment. During retrieval, the participants re-entered the environment, which was now missing either the necessary object or shadow cues and had to navigate to where they thought the target should be, confirming its location with a button press. The environment could be re-entered from one of four quadrants corresponding to a 0-, 90-, 180-, or 270-degree rotation from the initial starting position during encoding. Two additional elements were included in the task to elucidate performance differences between groups. First, the experiment consisted of “shadow” or “objects” blocks that informed the participants what type of information would most likely be available during retrieval—two objects without shadows (object trial) or one object with a shadow (shadow trial). Note that this was the minimal amount of information necessary to reorient in the environment. In each block, 70% of the trials were congruent (in accordance with the block); the remaining trials were switch trials—the unexpected, block-incongruent trials during retrieval. Second, before the start of each trial, participants saw a color cue informing them which of the three columns present during encoding would also be present in the retrieval phase of the trial. This meant that in object trials, this and one other column would be visible during retrieval. In shadow trials, the color cue designated the one column that would be available for reorientation in the environment. This was done to maintain distinct encoding of object locations and shadow directions, preventing shadows from being encoded on top of available object locations. Additionally, this served to balance the difficulty of the trials, given that judging distance might be considered easier in object trials due to the presence of the second column. Each block also included baseline trials matched for the block-congruent information during retrieval but did not require the participant to remember the location of the target. In these trials, the color cue was replaced by the text “no memory”. This meant that during retrieval, the target would be visible. The participants simply had to navigate towards it and stand on top of the target.

The OiS task was programmed in Blender, an open-source 3D content creation suite (The Blender Foundation Amsterdam, The Netherlands; www.blender.org, accessed 29 March 2011). Each trial began with the presentation of the color cue (1000 ms), followed by a blank screen (1000 ms) before the encoding phase started. The encoding phase had a maximum duration of 10 s. If the participant failed to navigate to the target object within the allocated time, a message appeared on the screen indicating that time had run out and a new trial began. This meant that if the participant did not complete the encoding phase, the corresponding retrieval phase was not shown. A fixation cross was presented for 4000 ms before the subsequent retrieval phase. During retrieval, 10 s were allocated for completing the task. If the participant failed to succeed in this, the trial stopped and continued to the next one. A fixation cross was shown (5000 ms) before the start of each new trial.

Conditions were counterbalanced in respect to a number of aspects. The distance from the starting point to the target during encoding and retrieval was kept equal between the objects and shadow conditions. The starting quarter in encoding and retrieval was counterbalanced across trials in addition to the amount of rotation (0, 90, 180, or 270 degrees) between the encoding starting quarter and that of retrieval. The color of the column closest to the target was also balanced across conditions. Furthermore, within the object trials, the availability during retrieval of the column closest to the target was counterbalanced, as well as the color of the column available during retrieval. Finally, an additional restriction was made to ensure that all columns should be visible upon entering the environment in the encoding and retrieval phases (i.e., no column could be occluded from view).

In order to navigate in the environment, participants used a button box with four buttons for the right hand to maneuver left (index finger), forward (middle finger), right (ring finger), and backwards (pinky finger). A separate button was used for the left index finger to confirm the placement of the target during retrieval. This was also necessary for “no memory” (baseline) trials, where the participant had to confirm that they were standing on top of the target.

### 2.3. Procedure

The experiment was divided into three sessions (2 behavioral, 1 scanning) all conducted on the same day, one after the other. With breaks between and during sessions, the entire experiment lasted 4 h (see Figure 2).

#### 2.3.1. Behavioral Session I

The experiment started with a behavioral session lasting about 1 h. The first task that was given to the participants was the Flanker Task, which lasted approximately 5 min. This was followed by the presentation of the instructions for the OiS task, which contained 5 practice trials. Next, participants completed the training version of the OiS task inside a dummy scanner (a simulation scanner that lacks the actual MRI scanner’s magnetic field and loud noise). The training version was identical to the actual task but shorter in duration. There were 4 blocks (2 Sun and 2 Objects blocks), each with 14 trials. The type of block that participants started with was randomized; the following blocks always alternated from one to the other so that two of the same blocks would never follow each other. Each block consisted of 7 congruent, 3 incongruent, and 4 no-memory trials. The total duration of the OiS training task (including instructions and familiarization with the dummy scanner) was approximately 40 min.

The OiS task training was conducted for two important reasons. First, the use of the dummy scanner ensured that all participants were familiar with the fMRI setup of the experiment and could perform the task in the scanner settings. Second, training the task beforehand eliminated the risk of confounding performance results with individual learning curves. Both measures ensured that participants were familiar with the task and the experimental setting before actual testing began and ruled out any potential confounding effects that might have been present between groups due to the amount of exposure to previous fMRI experiments.

#### 2.3.2. Scanning Session

The scanner session was divided into two runs. A total of 10 blocks (5 per run) of the OiS task were presented. As in the training phase, each block contained 14 trials (7 congruent, 3 incongruent, 4 no memory); 140 trials were presented in the entire experiment. The first block type was randomized among participants, with the following block types alternating with each other. Each run lasted approximately 35 min with a 5–10 min break in between the two runs. After the fMRI scanning, two structural scans were taken—a T1-weighted anatomical scan lasting 5 min and 21 s and a diffusion-weighted (DTI) scan lasting 10 min and 7 s.

#### 2.3.3. Behavioral Session II

The second behavioral session followed the scanning procedure and lasted 40–60 min. During this part of the experiment the participants filled out the SBSOD Questionnaire, Wayfinding Strategy Questionnaire, and Spatial Anxiety Scale, which took about 10 min to complete. Next, they were given two spatial tasks: the Mental Rotation Test (about 10 min) and the Eight-Arm Task (about 10 min). After these tests, the bilingual group additionally took the DIALANG to assess their English proficiency. This last test took 15–20 min to complete. All language-based tasks (questionnaires, DIALANG) were administered after the OiS task to ensure that the task was kept on as nonverbal a level as possible, preventing any carry-over effects of verbal material on the experimental task itself. Due to the long length of the whole experiment, the Language Background Questionnaire was filled out by the participants a few days before coming in for testing via an online link.

### 2.4. Functional MRI Data Acquisition and Analysis

The data was acquired on a Siemens 3 Tesla MAGNETOM Trio MRI scanner (Siemens Medical System, Erlangen, Germany) using a 32-channel coil. The acquisition of the functional scans was performed with a multiecho echo-planar imaging (EPI) sequence (TR = 2390 ms; TE = 9.4, 21.2, 33, 45, 57 ms; flip angle = 90°; 31 slices, 3 mm thick, FoV = 212 mm). This type of parallel acquisition sequence for functional images reduces motion and susceptibility artifacts (Poser et al., 2006) [75]. Structural T1 images were acquired using an MPRAGE sequence (TR = 2300 ms, TE = 3.03 ms, 192 sagittal slices, 1 mm thick, FoV = 256 mm).

Functional data were preprocessed and analyzed using the Matlab toolbox SPM8 (Statistical Parametric Mapping, London, UK). The first 4 volumes were discarded to control for T1 equilibration effects. The multiecho sequence acquired one volume for each of the five echoes at every time point. The first echoes were corrected for motion artifacts and the calculated corrections were applied to the remaining echoes of the same volume. The first 26 time points, acquired before the actual experiment started, were used as the weight calculation volumes (using the temporal signal-to-noise ratio for every voxel for the weight calculation) for the echo combination. Subsequently, all five echoes were combined into a single volume. To correct for time differences in the acquisition of slices within a volume, the time courses of each voxel were realigned to that of slice 16. The images were then coregistered to the participants’ anatomical scans, normalized to a standard template in MNI space, and smoothed with an 8 mm FWHM Gaussian kernel.

The resulting fMRI time series were analyzed in an event-related design within the general linear model (GLM). The time series of each condition (congruent objects encoding, congruent shadow encoding, no-memory encoding, congruent objects retrieval, congruent shadow retrieval, incongruent objects retrieval, incongruent shadow retrieval, no-memory retrieval) was convolved with a canonical hemodynamic response function and used as a regressor in the SPM multiple regression analysis. Events were time-locked to when the subjects first entered the environment in the encoding and retrieval phase of each trial. Fixation crosses and missed trials were also modeled. In addition, 6 realignment parameters were entered as effects of no interest.

Contrast images of the effects of interest (congruent objects encoding, congruent shadow encoding, congruent objects retrieval, congruent shadow retrieval, incongruent objects retrieval, incongruent shadow retrieval) were generated per subject. In addition, contrast images comparing the encoding conditions to the corresponding no-memory conditions and the retrieval conditions to the corresponding no-memory conditions were created. The first-level contrasts were entered into a random effects analysis.

Two separate types of analyses were conducted—one focusing on the task-specific activation (regardless of group) and the other on the bilingual–monolingual group differences. For the first type of analysis, first-level contrasts comparing each condition (objects or shadow) with the corresponding no-memory trials in the encoding or retrieval phase were used. A one-sample *t*-test was conducted to look for brain activation that was significantly different from zero on the group level between each condition and the no-memory trials. This analysis was carried out in the same way as that conducted in the original study by Baumann and colleagues [44].

For the main analysis of interest, i.e., focusing on group differences, three design matrixes were created in a 2 × 2 full factorial design for each of the following: congruent encoding, congruent retrieval, and incongruent retrieval. The congruent encoding design matrix consisted of two factors, each with two levels: group (bilingualism, monolingualism) and condition (objects, shadow). The condition contrast images were baseline corrected for differences in visual input among trials (first-level contrasts comparing the objects and shadow trials with no-memory trials were used). This was done to eliminate low-level visual activity resulting from focusing on particular aspects of the environment. An independent sample *t*-test was performed between bilinguals and monolinguals to look for significant differences between the groups in the objects and shadow conditions. Additional independent sample *t*-tests were performed for the effect of group on encoding. The analyses for the congruent and incongruent retrieval design matrix were performed in the same way as for the former matrix. The condition contrast images were not baseline-corrected in this case as the no-memory trials in the retrieval phase could not be perfectly matched for visual input with the other trials (the target was always visible in the environment). The conditions used in the incongruent design matrix were objects or shadow switch trials—trials that had a block-incongruent retrieval phase.

In order to eliminate group differences in activation based on general spatial abilities and proficiency in the experimental task, a set of five covariates were included as regressors of no interest in the aforementioned design matrixes. The scores of the spatial questionnaires and tasks were analyzed with a two-sample *t*-test for differences between bilinguals and monolinguals. This analysis yielded significant differences between groups for the Spatial Anxiety Scale and the Wayfinding Strategy Questionnaire (details of results can be found in the “Behavioral Results” section). The scores of these questionnaires were entered into the matrix as 3 regressors: anxiety, survey strategy, route strategy. Additionally, 2 regressors for performance (average error over trials) on congruent objects and shadow trials were included to control for performance-specific activation. In the case of the incongruent retrieval design matrix, these 2 regressors were the average error on objects and shadow switch trials.

The threshold statistics for clusters with *p* < 0.001 (uncorrected for the whole brain volume) were determined to be significant at *p* < 0.05 (corrected). For region of interest analyses (ROI), a small volume correction (SVC) was applied to the a priori defined regions. These regions consisted of known spatial brain areas: the bilateral hippocampus, left caudate nucleus, and bilateral parahippocampal gyrus [43,51,54]. We specifically chose the left caudate nucleus due to its additional role in monitoring and controlling language use in bilinguals [68]. Region masks were taken from the Anatomical Atlas Library (AAL) and were anatomically defined on a single brain matched to the MNI template [76]. 

## 3. Results

### 3.1. Behavioral Experiment

For all behavioral analyses, results were considered significant if the *p* threshold was below 0.05 two-tailed.

#### 3.1.1. Spatial Questionnaires and Tasks

Data from each questionnaire and task were analyzed with an independent samples *t*-test. Bilinguals (M = 66.5, SD = 12.1) did not differ from monolinguals (M = 65.5, SD = 15.3) in their scores on the SBSOD Questionnaire, t(27) = 0.2, *p* = 0.841. Data from the Wayfinding Strategy Questionnaire were analyzed separately for the survey scale and the route scale. Bilinguals (M = 2.69, SD = 0.60) and monolinguals (M = 3, SD = 0.68) did not differ in how much they used the survey strategy, t(27) = 1.27, *p*= 0.215, but did differ for the route strategy, t(27) = 2.49, *p* = 0.019, with monolinguals (M = 3.70, SD = 0.52) being more likely to use it than bilinguals (M = 3.07, SD = 0.81). Data from the Spatial Anxiety Scale showed that bilinguals (M = 2.12, SD = 0.61) had significantly lower scores than monolinguals (M = 2.65, SD = 0.54), t(27) = 2.52, *p* = 0.018. Bilinguals (M = 12.67, SD = 4.03) scored significantly higher on the MRT than monolinguals (M = 9.36, SD = 4.54), t(27) = 2.07, *p* = 0.048. Bilinguals (M = 1.07, SD = 1.49) and monolinguals (M = 1.71, SD = 1.38) did not significantly differ in the number of errors they made in the probe trial of the eight-arm task, t(27) = 1.21, *p* = 0.236.

#### 3.1.2. Flanker Task

The average duration of each trial of the flanker task was 459.72 ms, but individual trial durations varied from 338.15 to 713.65 ms for the entire participant group. Data from 1 participant who did not correctly complete the task (responded to the flankers instead of the target direction) were excluded from the analysis. Accuracy in the task ranged from 97.3 percent to 100 percent correct responses. A Group (2) by Condition (3) repeated analysis of variance of error rates did not reveal a significant effect of group, F(1, 26) = 3.68, *p* = 0.066, η^2^ = 124, or condition, F(2, 26) = 0.5, *p* = 0.61, η^2^ = 0.019. There was also no significant interaction effect between group × condition, F(2, 26) = 1.66, *p* = 0.201, η^2^ = 0.06.

Response times were analyzed with a Group (2) by Condition (3) repeated analysis of variance, which indicated a significant main effect of group, F(1,26) = 13.74, *p* < 0.001, η^2^ = 0.346 with bilinguals responding faster than monolinguals in all trial types. There was also a significant main effect of condition F(1.17, 26) = 110.3, *p* < 0.001, η^2^ = 0.809. Post hoc paired sample *t*-tests revealed that response times for incongruent trials were significantly slower than for congruent trials, t(27) = 10.28, *p* < 0.001, and neutral trials, t(27) = 10.27, *p* < 0.001; congruent and neutral trials did not differ, t(27) = 0.83, *p* = 0.416. The group × condition interaction effect was not significant, F(1.17,26) = 3.14, *p* = 0.081, η^2^ = 0.108 (for means see Appendix A).

#### 3.1.3. Objects in Space Task

##### Training

Data from the training phase of the OiS task were analyzed regarding performance errors, which were measured as the distance from the target’s location in encoding to where it was placed by the participant during retrieval. A Group (2) by Block (2) by Congruency (2) repeated analysis of variance revealed a significant three-way interaction effect, F(1,27) = 7.16, *p* = 0.013, η^2^ = 0.21. There were also significant interaction effects of block × congruency, F(1, 27) = 22.02, *p* < 0.001, η^2^ = 0.449, group × congruency, F(1, 27) = 6.51 *p* = 0.017, η^2^ = 0.194, and significant main effects of congruency, F(1, 27) = 53.26, *p* < 0.001, η^2^ = 0.664 and block, F(1, 27) = 13.52, *p* = 0.001, η^2^ = 0.334. The interaction effect group × block, F(1, 27) = 3.6, *p* = 0.069, η^2^ = 0.117 was not significant, as well as the main effect of group, F(1, 27) = 0.11, *p* = 0.744, η^2^ = 0.004. To investigate the three-way interaction, the data were reanalyzed with two separate analyses of variance models.

Two analyses of variance models with a Group (2) by Congruency (2) design were created for each block condition. For the shadow block, there was no significant group congruency interaction effect, F(1, 27) = 0.081, *p* = 0.778, η^2^ = 0.003 nor the main effect of group, F(1, 27) = 1.99, *p* = 0.170, η^2^ = 0.069 or congruency, F(1, 27) = 3.39, *p* = 0.076, η^2^ = 0.112. For the objects block, there was a significant main effect of congruency, F(1, 27) = 53.012, *p* < 0.001, η^2^ = 0.663 with congruent trials (M = 23.7, SD = 8.67) having less error than incongruent, as well as a significant group × congruency interaction effect, F(1, 27) = 10.267, *p* = 0.003, η^2^ = 0.275, indicating a stronger congruency effect for bilinguals than monolinguals. The main effect of group was not significant, F(1, 27) = 0.584, *p* = 0.451, η^2^ = 0.021.

##### Main Task (fMRI)

Data from the fMRI OiS task were analyzed for performance errors in the retrieval phase and the time it took to complete the encoding and retrieval part of each trial. Performance errors were analyzed in a Group (2) by Block (2) by Congruency (2) repeated analysis of variance. There was a significant main effect of block F(1, 27) = 13, *p* = 0.001, η^2^ = 0.325 and congruency F(1, 27) = 8.4, *p* = 0.007, η^2^ = 0.237 but not group, F(1, 27) = 0.954, *p* = 0.337, η^2^ = 0.034. There were significant interaction effects for group × block (F,27) = 4.586, *p* = 0.041, η^2^ = 0.145 and group × congruency, F(1, 27) = 18.3, *p* = < 0.001, η^2^ = 0.404. The block × congruency interaction effect, F(1, 27) = 2.161, *p* = 0.153, η^2^ = 0.074, and the three-way interaction effect, F(1, 27) = 1.42, *p* = 0.244, η^2^ = 0.05 were not significant. The mean performance errors for each condition are presented per group in Figure 3.

Post hoc independent and paired sample *t*-tests showed that bilinguals had lower errors in congruent trials than monolinguals, t(27) = 2.23, *p* = 0.034, but did not differ in their performance in incongruent trials, t(27) = 0.53, *p* = 0.60). Bilinguals had significantly lower error scores in congruent than incongruent trials, t(14) = 4.98, *p* < 0.001, but monolinguals did not significantly differ in their performance, t(13) = 1, *p* = 0.335. Bilinguals and monolinguals did not significantly differ in performance error in shadow, t(27) = 1.49, *p* = 0.147, or object blocks, t(27) = 0.355, *p* = 0.725, but bilinguals had higher error scores in object blocks than shadow blocks, t(14) = 3.55, *p* = 0.003, whereas monolinguals did not, t(13) = 1.3, *p* = 0.216.

The amount of time it took to complete the encoding phase of each trial was analyzed with a Group (2) by Block (2) by Congruency (2) repeated analysis of variance. There was a significant main effect of block, F(1, 27) = 11.46, *p* = 0.002, η^2^ = 0.298 and group × block interaction, F(1, 27) = 7.94, *p* = 0.009, η^2^ = 0.227. There was no significant main effect of group, F(1, 27) = 3.467, *p* = 0.074, η^2^ = 0.114 or congruency F(1, 27) = 0.42, *p* = 0.525, η^2^ = 0.015. There were also no significant group × congruency, F(1, 27) = 0.30, *p* = 0.587, η^2^ = 0.011, block × congruency, F(1, 27) = 0.30, *p* = 0.586, η^2^ = 0.011, and three-way interaction effects, F(1, 27) = 2.13, *p* = 0.156, η^2^ = 0.073. Post-hoc independent samples *t*-tests showed that bilinguals took more time in object trials than monolinguals, t(27) = 2.66, *p* = 0.013, but there was no significant difference in shadow trials, t(27) = 0.84, *p* = 0.409.

The amount of time it took to complete the retrieval phase of each trial was analyzed with a Group (2) by Block (2) by Congruency (2) repeated analysis of variance. There was no main effect of group, F(1,27) = 0.02, *p* = 0.898, η^2^ = 0.001, block, F(1,27) = 3.82, *p* = 0.061, η^2^ = 0.124 or congruency, F(1,27) = 4.01, *p* = 0.055, η^2^ = 0.129. There was a significant interaction effect of block × congruency, F(1,27) = 12.75, *p* < 0.001, η^2^ = 0.647, indicating that congruency had a stronger effect on the shadow block than the objects block. The remaining interaction effects were not significant for group × block, F(1, 27) = 0.07, *p* = 0.801, η^2^ = 0.002, group × congruency, F(1,27) = 1.87, *p* = 0.183, η^2^ = 0.065, and the three-way interaction effect, F(1,27) = 1.14, *p* = 0.295, η^2^ = 0.04.

We conducted separate Group (2) by Block (2) repeated analyses of variance for no-memory trials as they were not part of the experimental conditions and did not recruit any of the executive and spatial processes investigated in the study—they were used solely for the purpose of baseline correction in the fMRI analysis. For performance errors, there was no significant main effect of group, F(1, 27) = 0.006, *p* = 0.941, η^2^ < 0.001, block, F(1,27) = 0.148, *p* = 0.703, η^2^ = 0.005 or two-way interaction effect, F(1,27) = 0.543, *p* = 0.467, η^2^ = 0.02. For the time to complete the encoding phase, there was a main effect of group, F(1,27) = 8.68, *p* = 0.007, η^2^ = 0.243, with monolinguals taking more time to complete these trials than bilinguals. There was no main effect of block, F(1,27) = 0.67, *p* = 0.42, η^2^ = 0.024 or two-way interaction effect, F(1,27) = 1.02, *p* = 0.321 η^2^ = 0.036. For time to complete the retrieval phase, there was no significant main effect of group, F(1,27) = 2.89, *p* = 0.101, η^2^ = 0.097, block, F(1,27) = 1.17, *p* = 0.288, η^2^ = 0.042 and no two-way interaction effect, F(1,27) = 1.01, *p* = 0.323, η^2^ = 0.036.

Finally, a paired samples *t*-test on the total performance errors from the fMRI experiment and the training phase showed that there was a significant difference between the two parts of the task, with participants making fewer errors in the fMRI task (M = 78.58, SD = 20.65) than the training (M = 124.64, SD =27.14), t(28) = 11.94, *p* < 0.001.

### 3.2. Functional MRI Experiment

Analyses dedicated to the neural correlates of spatial cue use in a larger sample that completed this experiment can be found in [74]. Here, we focus on the direct comparison between bilinguals and monolinguals.

#### 3.2.1. Encoding

A contrast comparing bilinguals and monolinguals in encoding phases of the trials revealed significant activations for bilinguals in three large clusters covering the (1) right middle temporal gyrus and stretching bilaterally to the fusiform gyrus, (2) the opercular part of the right inferior frontal gyrus extending to the precentral gyrus, and (3) the left inferior temporal gyrus also covering the orbital part of the inferior frontal gyrus and temporal pole (Figure 4I). We found additional clusters in the left postcentral gyrus extending to the precentral gyrus and the triangular part of the inferior frontal gyrus, bilaterally in the SMA extending to the left superior medial frontal gyrus, the left inferior parietal cortex extending through the supramarginal gyrus to the angular gyrus, the right supramarginal gyrus, the left middle temporal gyrus, and in the right superior frontal and middle gyri (covering the dorsolateral prefrontal cortex (BA 9)). The reverse contrast did not reveal any significant activation in monolinguals.

On object trials, bilinguals had more activity in the left inferior parietal cortex and a cluster stretching from the left inferior temporal gyrus to the temporal pole. In the right hemisphere, we found a cluster in the angular gyrus that stretched to the middle occipital gyrus, as well as a cluster in the middle temporal gyrus and fusiform gyrus, covering also part of the right cerebellum. A similar contrast for the shadow trials revealed greater bilingual activations in the right lingual gyrus that stretched bilaterally to the fusiform gyrus. Other clusters were located in the right calcarine gyrus, extending to the cuneus, in the right precentral gyrus, in the right opercular part of the inferior frontal gyrus, and in the right middle occipital gyrus. Finally, we found a left hemisphere cluster in the orbital part of the inferior frontal gyrus. Reverse contrasts for objects and shadow trials did not reveal significant activation.

ROI analyses on the bilateral hippocampus, left caudate nucleus, and bilateral parahippocampal gyrus did not reveal any significant activation in the group effect of encoding, as well as for separate contrasts for objects and shadow trials.

For all significant activations in the encoding phase, see Appendix A.

#### 3.2.2. Retrieval—Congruent Trials

A contrast for the positive effect of bilingualism on retrieval revealed activations in the right superior frontal gyrus, superior medial and middle frontal gyri (encompassing the region of the dorsolateral prefrontal cortex, BA 9, 46). We found additional clusters in the right hemisphere in the cerebellum extending to the fusiform gyrus, superior orbitofrontal gyrus also covering the superior frontal gyrus, and the rolandic operculum stretching to the superior temporal gyrus. We also found a left superior temporal cluster as well as a cluster covering the left caudate nucleus stretching to the superior orbital frontal gyrus (Figure 4(IIA)). We did not find significant clusters of activation in the reverse contrast.

A group comparison of object trials did not reveal any significant activation. In a similar contrast for shadow trials, we found stronger activation for bilinguals in the superior frontal gyrus stretching through the dorsolateral prefrontal cortex (superior medial and middle frontal gyri). The reverse contrast for objects and shadow trials did not reveal any significant activation.

ROI analyses revealed significant activations for the positive effect of bilingualism in the right hippocampus and bilateral parahippocampal gyrus. In object and shadow trials, bilinguals showed greater activation in the left caudate nucleus compared to monolinguals. Additionally, in shadow trials, bilinguals activated the left parahippocampal gyrus. None of the reverse contrasts comparing monolinguals with bilinguals revealed significant clusters of activation. For all significant activations in the retrieval phase of congruent trials, see Appendix A.

#### 3.2.3. Retrieval—Incongruent Trials

A contrast for the positive effect of bilingualism on incongruent ‘switch’ trials revealed large clusters of activation covering the right superior frontal gyrus and stretching to the inferior frontal and superior medial gyri (encompassing the dorsolateral prefrontal cortex (BA 9)), and the right inferior temporal gyrus extending to the parahippocampal gyrus. We found additional activation in the left caudate nucleus that spread to the left anterior cingulated cortex and a cluster in the rolandic operculum and supramarginal gyrus. Finally, we found a cluster covering the left inferior orbital frontal gyrus and superior temporal pole (Figure 4(IIB)). The reverse contrast did not reveal any significant activation.

A group comparison of incongruent object trials, i.e., object trials with a switch in the retrieval phase, revealed larger activation in bilinguals in the right superior and superior medial frontal gyri. This activation included the dorsolateral prefrontal cortex (BA 9). We performed the same contrast for shadow switch trials and found an extensive cluster also covering the right superior and superior medial frontal gyri (also including BA 9). We additionally found an inferior temporal cluster in the right hemisphere. Reverse contrasts for objects and shadow trials did not reveal any significant activation.

ROI analyses revealed that bilinguals activated the right hippocampus significantly more than monolinguals in incongruent trials. When we compared bilinguals with monolinguals in object switch trials, we only found greater left caudate activation in bilinguals. The same contrast in shadow trials revealed more activation in bilinguals in the right hippocampus, left caudate nucleus, and left parahippocampal gyrus. For all significant activations in the retrieval phase of congruent trials, see Appendix A.

## 4. Discussion

In the current study, we aimed to investigate the extent of the bilingual executive control advantage in spatial cognition on both the behavioral and neural level. We used a three-dimensional virtual navigation task that required participants to use positional and directional spatial cues. We also collected data about participants’ spatial abilities and executive processes with a variety of additional questionnaires and tasks. We hypothesized that the bilingual advantage would manifest in task performance differences compared to monolinguals, as well as different neural activation patterns while completing the task.

The behavioral results of the spatial questionnaires point to differences between bilinguals and monolinguals in certain aspects of spatial cognition. Monolinguals were more likely than bilinguals to use the route strategy, referring to more step-by-step directions and local landmarks for wayfinding and had higher levels of spatial anxiety. These group differences are in line with previous findings, which showed a positive correlation between the use of the route strategy and spatial anxiety [69]. In the aforementioned study, worse sense of direction performance also correlated with higher spatial anxiety. In our study, monolinguals and bilinguals did not differ in their large-scale spatial abilities measured with the SBSOD questionnaire. The questionnaire was, however, an indirect measure of this ability, unlike the behavioral pointing task used in the previous experiment. Furthermore, the results of the eight-arm task suggest that there is no prevailing strategy used in navigation for bilinguals versus monolinguals. Both groups appear to use what Iaria and colleagues [43] and Bohbot and colleagues [52] call spatial and nonspatial (sequential) strategies to the same degree. Bilinguals were better on the MRT than monolinguals, suggesting that they may be better at mentally rotating figures and in line with previous studies [28,29], showing a bilingual advantage for mental rotation and imagery. 

We were able to replicate results from previous studies [13,14,77], showing a general bilingual advantage in the flanker task. Bilinguals were faster in both incongruent, congruent, as well as neutral trials. We did not, however, observe a reduced flanker effect for bilinguals compared to monolinguals, contrary to [13]. There is, however, an ongoing debate in the field concerning the bilingual advantage in executive control [1,78,79] and when it becomes apparent [80]. Bilinguals self-reported to use both English and Dutch often to daily and rated their language skills as good to very good. It is, therefore, a possibility that the language switching habits of the bilingual participants contributed to the significant switch effect in the Flanker task as well as to the performance in the OiS task. A recent study by Carter and colleagues [2] found that increased frequency of language switching corresponded to reduced interference effects in the flanker task, which, in turn, was beneficial for interference control. Of note, other studies have also used the Simon task instead of the Flanker task to demonstrate the bilingual advantage [12].

Bilinguals showed a significant switch effect, with incongruent trials having greater errors than congruent ones, whereas monolinguals did not. This was driven by better bilingual performance in congruent trials. Since performance in memory trials was the same for both groups, it is unlikely that bilinguals were better at maneuvering in the environment. This is further supported by the fact that bilinguals and monolinguals did not differ in their overall performance in the training version of the Objects in Space task for both block types, suggesting that the groups were matched in terms of baseline proficiency.

General task-specific results in the OiS task showed distinct neural correlates for the encoding and retrieval phases and block types. Objects and shadow trials activated specific spatial areas important for processing positional and directional spatial cues, in line with Baumann and colleagues [59] for a similar task design. Neural activity for a larger sample of participants completing this task is described by Wegman and colleagues [74]. Below, we discuss neural activation differences between bilinguals and monolinguals.

Strikingly, during the encoding phase, bilinguals, compared to monolinguals, activated similar regions to those identified by a quantitative meta-analysis study on the neural correlates of bilingual language switching [24], suggesting that bilinguals may tap into this network for solving complex or demanding tasks that require managing multiple representations. These regions include the left inferior frontal gyrus (IFG) and bilateral temporal regions, as well as the supramarginal gyrus—according to Abutalebi and Green [20] involved in bilingual language processing. The IFG has been suggested to better respond to task demands due to the necessity of managing two languages [81], and the left IFG has also been implicated in bilingual language use and proficiency [82]. During the encoding phase of the trials, participants had to recruit extensive mental resources in order to successfully complete the task. Consequently, the task was quite demanding and required good orientation and executive control abilities. In light of previous findings, it may be that bilinguals engaged an area that they use on a daily basis for attenuating language switch costs. The right IFG has also been associated with nonverbal bilingual performance [26]. The aforementioned study found that the right IFG was related to better performance in congruent trials in a flanker task. More right IFG activation in the OiS task may suggest more efficient encoding of expected spatial cues by bilinguals, reflected by their performance in these trials. Finally, the network of frontal and prefrontal areas, as well as the parietal cortex activated by bilinguals during the encoding phase, is also in line with the neural correlates suggested to subserve the alerting, orienting, and executive attentional networks [21].

The encoding phase was further analyzed with respect to object trials and shadow trials. During object trials, participants were to focus on the landmark-based or positional relations between the target and the colored columns in order to be able to later reorient in the environment. Bilinguals, compared to monolinguals, activated bilateral temporal regions, the left inferior parietal cortex, right middle occipital and angular gyri, right fusiform gyrus, and right cerebellum. The occipital–parietal areas and temporal regions also overlap with activations found in the task-specific contrast, suggesting that bilinguals engaged these regions more for additional spatial processing. The angular gyrus, which was present in both task-specific and group comparisons, has been suggested as an area that integrates all spatial aspects of the environment, combining external sensory information with internal mental representations [83]. Bilinguals additionally recruited the inferior parietal cortex, an area that has been associated with not only visuospatial perception [60], but orienting and attentional processes [21]. These results may suggest that bilinguals recruited a more extensive network for encoding landmark-based locations than monolinguals, specifically engaging the parietal region.

During shadow trials, the most relevant information for successful completion of the task was the orientation of the shadows that were cast from the column, enabling reorientation in the environment when no other cues were available. For these trials, bilinguals more than monolinguals activated the right lingual gyrus, calcarine gyrus, cuneus, middle occipital gyrus, precentral gyrus, bilateral fusiform gyrus and IFG. The activation of an extensive prefrontal network, especially the IFG, suggests bilingual-specific recruitment of areas involved in general cognitive control [81] and bilingual language switching [24]. Lastly, more activation in the fusiform gyrus and the lingual gyrus (which feeds into the parahippocampal gyrus)—areas relevant for spatial navigation [42,44], tentatively suggests that bilinguals may increase engagement of certain spatial areas, in addition to executive control regions, to facilitate better encoding of spatial information. Taken together, the striking difference between bilinguals and monolinguals during the encoding phase is enhanced activation of attentional networks as well as areas known to subserve bilingual language maintenance. It may be that increased activation of these areas facilitates better encoding of spatial information by maintaining both positional and directional spatial cues.

During the retrieval phase, participants entered the environment again and had to navigate to where they thought the target should be. During this phase, there was only one type of spatial cue available—columns or shadows—constituting the minimal amount of spatial information necessary to reorient in the environment. The retrieval phase could be either congruent with the block information or incongruent (constituting a switch trial). In the retrieval phase of congruent trials, bilinguals, compared to monolinguals, activated a similar network as in the encoding phase: bilateral frontal areas, bilateral temporal areas, the right cerebellum and the right fusiform gyrus. Additionally, bilinguals also recruited areas known to subserve landmark-related and world-based spatial cue use: the left caudate nucleus, right hippocampus, and bilateral parahippocampal gyrus. The caudate nucleus and hippocampus are thought to contribute in a noncompetitive way to route recognition [54,84]. The caudate nucleus has specifically been shown to code landmark-related spatial information [54] and positional local cues (Iaria et al., 2003) [43]. The hippocampus has been implicated in survey and boundary-related spatial cue processing [43,54]. The bilateral parahippocampal gyrus is activated for objects that are navigationally relevant [48], and its functional connectivity with the hippocampus is related to better self-reported navigation abilities [51]. Our results suggest that bilinguals had increased recruitment of a wide network of regions subserving spatial cue processing and navigation. The activation of the caudate nucleus is particularly interesting as it can underlie both spatial and cognitive control processes in our bilingual group. For example, the left caudate nucleus is specifically known to control language conflict [20,85]. Its robust activation in the retrieval phase of congruent trials for bilinguals suggests that it may be better ‘trained’ to meet various task demands and, hence, more readily engaged in such a task.

Incongruent trials in the retrieval phase produced increased activity in bilinguals in an extensive network subserving executive control and spatial processing. Bilinguals compared to monolinguals activated the DLPFC and left rostral–ventral ACC (rvACC). It may seem that the enhanced engagement of this network, especially the ACC, is contradictory to the properties of the executive control network attributed to bilingual conflict resolution and monitoring in other studies [20,27]. These studies point to a decrease in ACC activity for bilinguals compared to monolinguals as a result of tuning in this region through continual language switching. There is, however, a discrepancy regarding the exact region in question. Within the ACC, there is a functional division between the cognitive dACC and the emotional rvACC [86]. The meta-analysis conducted by Bush and colleagues [86] shows that the cognitive division is activated by executive control tasks, such as the Stroop task, as well as divided attention and complex response selection tasks. Emotional division is activated by tasks that have affective, emotional or provocative content. The region of the ACC that was more activated by the bilinguals in our study was rostral and, therefore, likely not directly related to the cognitive component. Bilinguals might either recruit different areas for dealing with conflict, such as the IFG or left caudate (which were also activated during the switch trials), or have a more proficient conflict resolution/monitoring ACC. The results of the task-specific activation further perpetuate this hypothesis. In the retrieval phase of incongruent trials, there was an increase of activity in the dACC, in line with the properties of the cognitive subdivision, and suggests that the switch trials did induce conflict. Bilinguals, however, activated the left caudate nucleus with activity spreading into the rACC.

It should be noted that there is an ongoing debate in the field with respect to the bilingual advantage in executive control [1,2,3,4,5,6,7,8,78,79]. Whereas some researchers show bilingual advantages in executive switch tasks such as the Flanker or Simon task, others find no such advantage [4,5,6,7,8]. Whether this is due to slight differences in how the tasks were used, e.g., differences in stimulus intervals or a failure to replicate, remains elusive. A recent study by Xia and colleagues [3] observed that types of conflicts involved in interference tasks, as well as language proficiency of mono- and bilinguals, differentially affect performance. Furthermore, it has recently been suggested that the presence and magnitude of the bilingual advantage may be dependent on individual and context-specific factors [80,87,88]. In this study, we use a navigation task that simulates real-world settings to investigate the bilingual advantage in more complex and demanding situations.

The generalizability of the presented findings is limited by the low sample size of the bilingual and monolingual groups. This should, however, be weighed against the specificity and careful characterization of both samples. While these findings should be taken tentatively and replicated in future studies, they provide initial insight into potential neural correlates of bilingual spatial processing in complex environments.

It could be argued that the increased engagement of the executive control network in bilinguals reflects the allocation of additional resources to meet task demands in order to match monolinguals’ performance. Monolinguals had a tendency for overall worse performance in both congruent and incongruent trials, with no switch costs. As such, they may not have experienced conflict in the same way as bilinguals did. The bilingual advantage present in congruent trials may have, therefore, necessitated the recruitment of additional resources to compensate for switch costs.

## 5. Conclusions

The results of this exploratory study point to a bilingual-specific recruitment of extensive networks encompassing both executive control and spatial regions. Bilinguals engaged multiple regions underlying both positional and directional spatial cue use in a navigation task while matching or exceeding monolinguals in performance. While the greater engagement of the executive control regions during encoding may be related to increased proficiency, its involvement during retrieval appears to have allowed for compensation of bilingual switch costs. In conclusion, this study suggests that the bilingual advantage often observed in some standard executive control tasks also holds in the spatial domain for spatial cue use.

## Figures and Tables

**Figure 1 brainsci-14-00134-f001:**
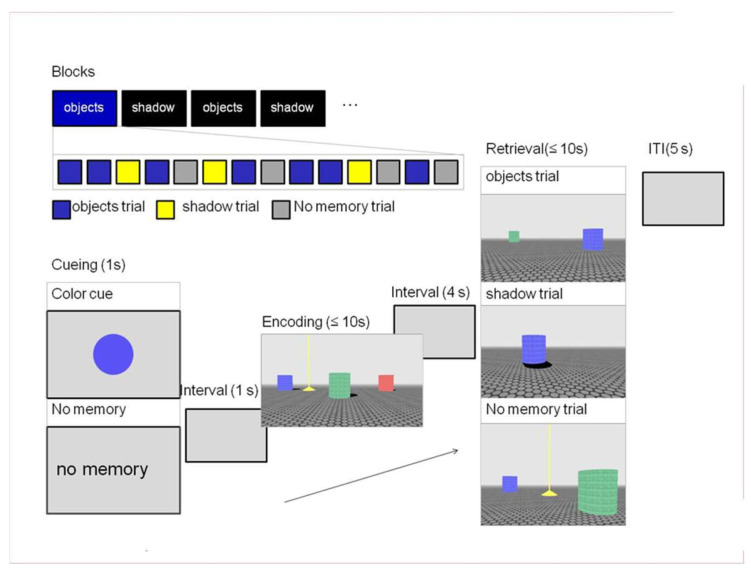
**Figure 1** is adapted from Wegman and colleagues [74]. The Objects in Space Task. Each trial consisted of an encoding and retrieval phase. Participants had to navigate toward a target (shown in yellow) that was visible during encoding but hidden in retrieval. During encoding, three object cues (columns in red, green and blue) with their shadows were visible. Before each trial, a color cue indicated which of the colored columns would be available during retrieval. During retrieval, either two objects without shadows (object trial) or one object with a shadow (shadow trial) were available. Participants were instructed to move to the target position that was shown during encoding. Only in baseline trials, indicated at the cue phase, was the target visible during retrieval.

**Figure 2 brainsci-14-00134-f002:**
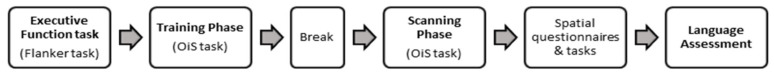
Flowchart shows the procedure of the experiment.

**Figure 3 brainsci-14-00134-f003:**
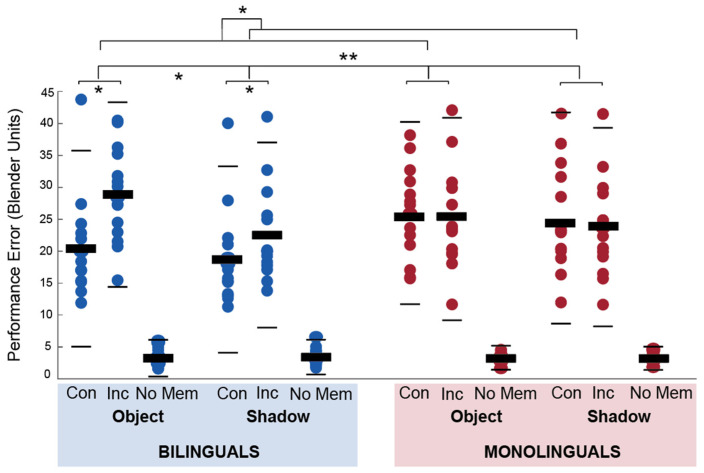
Objects in Space task mean performance errors per block, condition, and group. There were significant group × congruency and group × block interaction effects. Dots represent the average error per participant. Thick black bars indicate the mean, thin lines +/− 2 SD. Con, congruent; Inc, incongruent; No Mem, no-memory trials, ** *p* < 0.001, * *p* < 0.05.

**Figure 4 brainsci-14-00134-f004:**
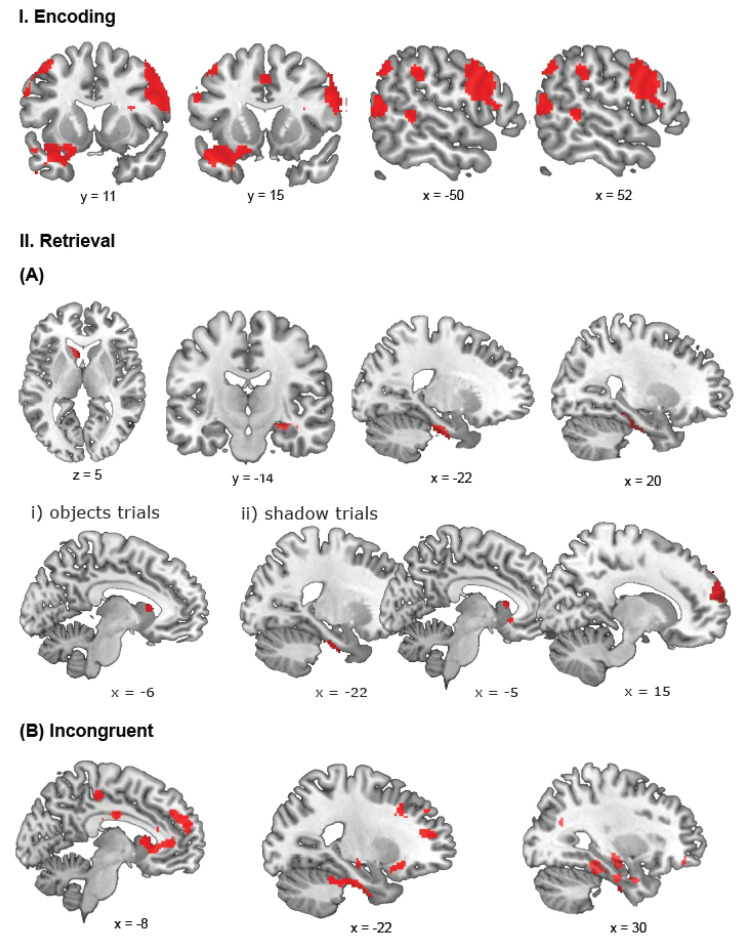
Task activation (in red) during the Objects in Space Task. During encoding, bilinguals activated temporal, parietal regions and inferior frontal regions more than monolinguals (**I**). During retrieval, bilinguals activated the left caudate nucleus, right hippocampus and bilateral parahippocampal gyrus more than monolinguals (**IIA**). Contrasts for specific trials showed that bilinguals activated the left caudate nucleus and parahippocampal gyrus in object trials (i) and right DLPFC, left parahippocampal and caudate regions in shadow trials (ii). Retrieval-related activation in incongruent ‘switch’ trials involved the ACC, frontal and inferior frontal regions, parahippocampal gyrus, hippocampus, and caudate nucleus (**IIB**). For visualization purposes, activation clusters are presented at a cluster-uncorrected threshold of *p* < 0.001.

## Data Availability

The data presented in this study are available on request from the corresponding author. The data are not publicly available because no written consent for public data sharing was obtained from participants.

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
