# Peer review of "Bilingual Spatial Cognition: Spatial Cue Use in Bilinguals and Monolinguals"

_brainsci, 2024, doi:10.3390/brainsci14020134_

Round 1

Reviewer 1 Report

Comments and Suggestions for Authors

Review Results

Manuscript entitled: Bilingual Spatial Cognition: Spatial Cue Use in Bilinguals and Monolinguals

This manuscript tried to explore the use of spatial cues in 15 bilinguals and 14 monolinguals while navigating in an open field virtual environment by using fMRI. The authors found that bilinguals may recruit neural networks known to subserve not only executive control processes, but also spatial strategies.

I really appreciated the authors to careful conduct the experiment and investigation. However, there are some minor concerns, as follows:

Part 2: Materials and Methods:

If the authors can add the flow chart of the study procedure, it will help the reader understand the whole experiment and concept.

Page 3, Line 122: The authors indicated that “Thirty-two healthy right-handed adults participated in the study.” How did the participants’ handedness dominant were assessed? By self-report?

The references were a bit old. From 79 citations, only two recent references [33] and [39] in 2022 was cited while the were more than 10 years. If the authors can revise this manuscript with more recent relevant references especially those recent 5-10 years, it would make this manuscript become stronger (even though the protocol was approved in 2010).

More importantly, the authors mentioned that “a protocol approved by the local ethics committee (CMO region Arnhem–Nijmegen, The Netherlands, approval code CMO2001/095, December 2010)”. Actually, this approved protocol was issued 14 years ago. I become curious when this experiment was conducted.

Author Response

This manuscript tried to explore the use of spatial cues in 15 bilinguals and 14 monolinguals while navigating in an open field virtual environment by using fMRI. The authors found that bilinguals may recruit neural networks known to subserve not only executive control processes, but also spatial strategies.

I really appreciated the authors to careful conduct the experiment and investigation. However, there are some minor concerns, as follows:

Part 2: Materials and Methods:

If the authors can add the flow chart of the study procedure, it will help the reader understand the whole experiment and concept.

We agree, we have now added a flow chart as Figure 2 on page 8.

Page 3, Line 122: The authors indicated that “Thirty-two healthy right-handed adults participated in the study.” How did the participants’ handedness dominant were assessed? By self-report?

The participants indicated their handedness by self-report. We have now included this in the participant section on page 3, lines 125, 126: “Thirty-two healthy right-handed adults participated in the study, who indicated their handedness via self-report.”

The references were a bit old. From 79 citations, only two recent references [33] and [39] in 2022 was cited while the were more than 10 years. If the authors can revise this manuscript with more recent relevant references especially those recent 5-10 years, it would make this manuscript become stronger (even though the protocol was approved in 2010).

We have cited a number of more recent publications that contribute to the bilingual and monolingual performance in executive functioning.

Carter, F.; DeLuca, V.; Segaert, K.; Mazaheri, A.; Krottet, A. Functional neural architecture of cognitive control mediates the relationship between individual differences in bilingual experience and behaviour. NeuroImage 2023, 273, 120085. doi: 10.1016/j.neuroimage.2023.120085.

Xia, L.; Bak, T.H.; Sorace, A.; Vega-Mendoza, M. Interference suppression in bilingualism: Stimulus-Stimulus vs. Stimulus-Response conflict. Bilingualism: Language and Cognition, 2022, 25(2), 256-268 doi.10.1017/S1366728921000304

Paap, K.R. The bilingual advantage debate: Quantity and quality of the evidence. In J. W. Schwieter (Ed). The Handbook of Neuroscience of Multilingualism 2019 (pp. 701-735). London: Wiley-Blackwell.

Paap, K.R.; Myuz, H.; Anders-Jefferson, R.; Mason, L.; & Zimiga, B. On the ambiguity regarding the relationship between sequential congruency effects, bilingual advantages in cognitive control, and the disengagement of attention. AIMS neuroscience 2019, 6(4), 282.

Paap, K.R.; Mason, L.A.; Zimiga, B.M.; Ayala-Silva, Y.; Frost, M.M.; Gonzalez, M.; Primero, L. Other Language Proficiency Predicts Unique Variance in Verbal Fluency Not Accounted for Directly by Target Language Proficiency: Cross-Language Interference?. Brain Sci 2019, 9(8), 175.

Paap, K.R.; Anders-Jefferson, R.; Mikulinsky, R.; Masuda, S.; Mason, L. On the encapsulation of bilingual language control. Journal of Memory and Language 2019, 105, 76-92.

Paap, K.R.; Myuz, H.A.; Anders, R.T.; Bockelman, M.F.; Mikulinsky R.; Sawi, O.M. No compelling evidence for a bilingual advantage in switching or that frequent language switching reduces switch cost, J. Cogn. Psy., 2016 DOI: 10.1080/20445911.2016.1248436

More importantly, the authors mentioned that “a protocol approved by the local ethics committee (CMO region Arnhem–Nijmegen, The Netherlands, approval code CMO2001/095, December 2010)”. Actually, this approved protocol was issued 14 years ago. I become curious when this experiment was conducted.

The experiment was conducted in 2013. An earlier publication from Wegman and colleagues from 2014 shows the neural activity for a larger sample of participants who completed these tasks. In this publication the language background of the participants was not analysed.

Reviewer 2 Report

Comments and Suggestions for Authors

The manuscript overall is well-written. The authors provided further bilingual advantage evidence for a spatial task. Neural correlates and behavioral measurements were also included. The methods and results were also clearly presented. I have some comments. 

1. As authors themselves admit, the sample size was relatively small, so I would like to see the overal effect sizes for significant findings. 

2. The authors mentioned the bilingual advantage debate, so I would like to see some discussion on this topic in depth. I did not see the authors cited the work of K. Paap.  His work on this topic is a valuable addition to the work of E. Bialystok. 

3. Why flanker task was adopted, because we know simon task was also widely used. 

  Comments on the Quality of English Language

The authors could edit the manuscript a little bit. Some paragraphs are not divided properly. 

Author Response

The manuscript overall is well-written. The authors provided further bilingual advantage evidence for a spatial task. Neural correlates and behavioral measurements were also included. The methods and results were also clearly presented. I have some comments. 

  1. As authors themselves admit, the sample size was relatively small, so I would like to see the overal effect sizes for significant findings. 

We have added the effect sizes (h2) for the behavioral effects of the OiS task and the Flanker task. With respect to the neural findings, the cluster size of the main effects can be used as an indicator of effect size. The cluster forming threshold was p<.001 uncorrected which indicates a large cluster of neural activity. Importantly, the main prefrontal findings for the effect of group on encoding trails are two very large clusters of k=1150 and k=1084 voxels. For retrieval congruent trials this is also a large prefrontal cluster of k=508 voxels and retrieval incongruent trials k=2339 voxels. The cluster sizes for all clusters are reported in the supplemental tables.

  1. The authors mentioned the bilingual advantage debate, so I would like to see some discussion on this topic in depth. I did not see the authors cited the work of K. Paap.  His work on this topic is a valuable addition to the work of E. Bialystok. 

Thank you, we agree that the work of K. Paap and colleagues is very valuable. We have now added a number of publications to the discussion (please see the six publications listed below). Furthermore, we have added the following part to the discussion on page 19, lines 856-866. These publications are also added to the Introduction as suggested by reviewer 1.

“It should be noted that there is an ongoing debate in the field with respect to the bi-lingual advantage in executive control [1-8,77,78]. Whereas some researchers show bilingual advantages in executive switch task such as the Flanker or Simon task, others find no such advantage [4-8]. Whether this is due to slight differences in how the tasks were used, e.g. differences in stimulus intervals, or a failure to replicate remains elusive. A recent study by Xia and colleagues [3] observed that types of conflicts involved in interference tasks as well as language proficiency of mono- and bilinguals differentially affect performance. Furthermore, it has recently been suggested that the presence and magnitude of the bilingual advantage may be dependent on individual and context-specific factors [79,85,86]. In this study, we use a navigation task that simulates real world settings to investigate the bilingual advantage in more complex and demanding situations.”

Added Literature from K. Paap and colleagues:

Paap, K.R. The bilingual advantage debate: Quantity and quality of the evidence. In J. W. Schwieter (Ed). The Handbook of Neuroscience of Multilingualism 2019 (pp. 701-735). London: Wiley-Blackwell.

Paap, K.R.; Myuz, H.; Anders-Jefferson, R.; Mason, L.; & Zimiga, B. On the ambiguity regarding the relationship between sequential congruency effects, bilingual advantages in cognitive control, and the disengagement of attention. AIMS neuroscience 2019, 6(4), 282.

Paap, K.R.; Mason, L.A.; Zimiga, B.M.; Ayala-Silva, Y.; Frost, M.M.; Gonzalez, M.; Primero, L. Other Language Proficiency Predicts Unique Variance in Verbal Fluency Not Accounted for Directly by Target Language Proficiency: Cross-Language Interference?. Brain Sci 2019, 9(8), 175.

Paap, K.R.; Anders-Jefferson, R.; Mikulinsky, R.; Masuda, S.; Mason, L. On the encapsulation of bilingual language control. Journal of Memory and Language 2019, 105, 76-92.

Paap, K.R.; Myuz, H.A.; Anders, R.T.; Bockelman, M.F.; Mikulinsky R.; Sawi, O.M. (2016): No compelling evidence for a bilingual advantage in switching or that frequent language switching reduces switch cost, J. Cogn. Psy., 2016 DOI: 10.1080/20445911.2016.1248436

  1. Why flanker task was adopted, because we know simon task was also widely used. 

We chose the flanker task in order to replicate results from previous studies, as we state on page 6, lines 268, 269, “The flanker task was used to verify that the bilingual group in our experiment could replicate results from previous studies (e.g., [1,13])”.  Additionally, the flanker task was chosen because it has been shown to differentiate bilinguals and monolinguals with respect to neural substrates of interference suppression (see Luk et al. 2010 ) and not response inhibition – the former process being more in line with the demands of the OiS task. We agree with the reviewer that the Simon task has also been used in the literature. We have now added the following statement to the discussion section on page 17, lines 743, 744: “Of note, other studies have also used the Simon task, instead of the Flanker task, to demonstrate the bilingual advantage [12].”

Reviewer 3 Report

Comments and Suggestions for Authors

Title: Bilingual Spatial Cognition: Spatial Cue Use in Bilinguals and Monolinguals

Overall assessment:

This paper examines whether bilinguals have a processing advantage over monolinguals in the spatial domain, using fMRI. The authors looked at network activation at encoding and retrieval while completing a spatial navigation task. They found that bilinguals showed more network activation than monolinguals during both encoding and retrieval of information.

The study and the writing are strong, and the topic seems relevant for the journal and the broader community. There is however one weakness to address: the bilinguals were not asked about their codeswitching habits although the authors conclude that the bilingual advantage found is likely due to continuous switching between languages.

This comment is further described below along with two relatively minor comments. I’d like to see this main question addressed.

Main comments:

-       A mention of meta-analyses and/or studies that find null results for a bilingualism advantage is presented in the discussion, but these would need to also be referenced early in the introduction, e.g. as a caveat in the second sentence, and in the 3rd paragraph, where references 5-9 are cited.

-       The last sentence of the conclusion sounds too strong considering that the bilingual advantage is not a reliable effect. It assumes there is a bilingual advantage at all times which is not what has been found in the literature. Perhaps add “in some EC tasks” or any other language that shows there is ambiguity in this area.

Methods

-       The bilingual participants were not asked about their language switching habits between English and Dutch. This seems to be a major weakness of the study as the discussion suggests that the bilingual advantage found in spatial orientation is a result of “continual language switching” (l. 821-2). We in fact do not know the extent to which bilinguals in this study engage in codeswitching, and how dense the codeswitching is between Dutch and English if present. Some further discussion to clarify this point or to address it as a limitation is needed.

Minor comments:

Line 703: There is an extra “to” in the sentence (“…to use to the route strategy”.)

Author Response

Overall assessment:

This paper examines whether bilinguals have a processing advantage over monolinguals in the spatial domain, using fMRI. The authors looked at network activation at encoding and retrieval while completing a spatial navigation task. They found that bilinguals showed more network activation than monolinguals during both encoding and retrieval of information.

The study and the writing are strong, and the topic seems relevant for the journal and the broader community. There is however one weakness to address: the bilinguals were not asked about their codeswitching habits although the authors conclude that the bilingual advantage found is likely due to continuous switching between languages.

This comment is further described below along with two relatively minor comments. I’d like to see this main question addressed.

 Main comments:

 -       A mention of meta-analyses and/or studies that find null results for a bilingualism advantage is presented in the discussion, but these would need to also be referenced early in the introduction, e.g. as a caveat in the second sentence, and in the 3rd paragraph, where references 5-9 are cited.

We have now adjusted the second sentence of the introduction and included a paragraph about the controversy regarding the bilingual advantages to the 3rd paragraph of the introduction. In addition, we cited new literature about the ongoing debate. We have also added a more in-depth part of the debate to the Discussion (page 19, lines 856-866) as suggested by reviewer 2.

Page 1, lines 26-29: “Many have argued that the use of a second language engages domain-general executive functions leading to a ‘bilingual advantage’ [1-3], whereas others find no evidence for a bilingual advantage in executive task switching [4-8].”

Page 2, lines 44-48: “Studies show that bilinguals compared to monolinguals have an advantage in nonverbal tasks recruiting inhibition, task switching, and selective attention processes [12-16]. This bilingual benefit is not always observed. There is an ongoing controversy in the field regarding the bilingual advantage in executive control [1-8].”

-       The last sentence of the conclusion sounds too strong considering that the bilingual advantage is not a reliable effect. It assumes there is a bilingual advantage at all times which is not what has been found in the literature. Perhaps add “in some EC tasks” or any other language that shows there is ambiguity in this area.

We agree, the bilingual advantage is not always observed. For clarity we have now rewritten the last sentence of the conclusion on page 20, lines 886-888. “In conclusion, this study suggests that the bilingual advantage often observed in some standard executive control tasks also holds in the spatial domain for spatial cue use.”

 Methods

 -       The bilingual participants were not asked about their language switching habits between English and Dutch. This seems to be a major weakness of the study as the discussion suggests that the bilingual advantage found in spatial orientation is a result of “continual language switching” (l. 821-2). We in fact do not know the extent to which bilinguals in this study engage in codeswitching, and how dense the codeswitching is between Dutch and English if present. Some further discussion to clarify this point or to address it as a limitation is needed.

All bilingual participants were required to use both English and Dutch. These participants filled in the L2 Language History Questionnaire. They reported using English often to daily and rated their general language fluency at least as good to very good. Ratings (means and SDs) from the Questionnaire are reported in 2.1.1 Language Background. We have highlighted these parts in the manuscript on page 4.

We have added a part on the bilingual language switching habits now in the discussion on page 17 (lines 737-744), which clarifies the switching habits of the bilingual group and the possible contribution to the Flanker task results:

Bilinguals self-reported to use both English and Dutch often to daily and rated their languages skills as good to very good. It is therefore a possibility that the language switching habits of the bilingual participants contributed to the significant switch effect in the Flanker task as well as to the performance in the OiS task. A recent study by Carter and colleagues [2] found that increased frequency of language switching corresponded to reduced interference effects in the flanker task which in turn was beneficial for interference control. Of note, other studies have also used the Simon task, instead of the Flanker task, to demonstrate the bilingual advantage [12].”

Minor comments:

Line 703: There is an extra “to” in the sentence (“…to use to the route strategy”.)

Thank you, we have changed the typo in this sentence.
